# Nodal Low-Grade B-Cell Lymphoma Co-Expressing CD5 and CD10 but Not CD23, IRTA1, or Cyclin D1: The Diagnostic Challenge of a Splenic Marginal Zone Lymphoma

**DOI:** 10.3390/diagnostics14060640

**Published:** 2024-03-18

**Authors:** Khin-Than Win, Yen-Chuan Hsieh, Hung-Chang Wu, Shih-Sung Chuang

**Affiliations:** 1Department of Pathology, Chi-Mei Medical Center, Tainan 710, Taiwan; 2Department of Clinical Pathology, Chi-Mei Medical Center, Tainan 710, Taiwan; 540003@mail.chimei.org.tw; 3Division of Hemato-Oncology, Department of Internal Medicine, Chi-Mei Medical Center, Tainan 710, Taiwan

**Keywords:** CD5, CD10, marginal zone lymphoma, phenotypic aberrancy, splenic marginal zone lymphoma, villous lymphocyte

## Abstract

The diagnosis of lymphoma is based on histopathological and immunophenotypical features. CD5 and CD10 are traditionally considered a T-cell antigen and a germinal center B-cell antigen, respectively. It is very unusual for a low-grade B-cell lymphoma (BCL) to co-express CD5 and CD10. Although the biologic basis or clinical significance of such co-expression is unclear, this rare event may pose a significant diagnostic challenge. Here, we report a case of a 63-year-old male presenting with bilateral cervical lymphadenopathy and lymphocytosis. Histologically, the nodal tumor was largely diffuse with neoplastic small atypical lymphocytes co-expressing CD5, CD10, and CD20, but not CD23 or cyclin D1. The leukemic cells in the peripheral blood exhibited hairy projections. Taking together the marked splenomegaly, involvement of lymph nodes, bone marrow, and peripheral blood, a final diagnosis of splenic marginal zone lymphoma (SMZL) was reached. The patient was alive with partial response for 10 months after immunochemotherapy. The dual expression of CD5 and CD10 is extremely unusual for low-grade BCL and may lead to an erroneous diagnosis. Integrating the findings into peripheral blood smear tests, flow cytometry, histopathology, imaging, and clinical features is mandatory to exclude other lymphoma types and to reach a correct diagnosis, particularly for a case with nodal presentation.

The diagnosis of lymphoma is based on histopathology and immunophenotyping. CD5 is traditionally considered a T-cell marker and is expressed in certain types of B-cell lymphomas (BCLs). Among various types of low-grade BCLs, CD5 is typically expressed in small lymphocytic lymphoma/chronic lymphocytic leukemia (SLL/CLL), together with CD23 [1]. The other B-cell lymphoma type with frequent CD5 expression is mantle cell lymphoma (MCL) [1]. CD10 is a marker of neutrophils and germinal center B-cells, and its expression in BCLs is usually a hint towards a germinal center cell origin, most frequently low-grade or so-called classic follicular lymphoma (cFL) [1,2]. 

Marginal zone lymphoma (MZL) is an indolent mature BCL, most commonly occurring in extranodal sites. When an MZL involves a lymph node, it might represent a primary nodal MZL (NMZL), or more commonly, an extranodal MZL (such as marginal zone lymphomas of mucosa-associated lymphoid tissue type (MALT) lymphoma or splenic MZL (SMZL)) with secondary nodal involvement. There is no specific phenotype of various MZLs, and the diagnosis relies on histopathology and immunophenotypic features with exclusion of other specific lymphoma types such as SLL/CLL, cFL, or MCL. BCLs co-expressing CD5 and CD10 are very unusual and may pose a diagnostic challenge.

A 63-year-old male patient presented with bilateral cervical masses in March 2023. There was no specific past medical history. He did not have fever or any other symptoms. Laboratory examination showed anemia, leukocytosis (WBC count at 51.4 × 10^3^/μL) with 33% abnormal lymphocytes exhibiting villous projections (absolute abnormal lymphocyte count at 16.7 × 10^3^/μL; Figure 1A), and a normal platelet count. Computed tomography (CT) scans of the neck revealed lymphadenopathy in bilateral cervical, axillary, and superior mediastinal regions. A right cervical lymphadenectomy was performed. We followed the guidelines of the Declaration of Helsinki. We obtained signed informed consent from the patient, and the study was approved by our Internal Review Board.

Flow cytometric immunophenotyping of the atypical lymphocytes in the peripheral blood showed a CD5+/CD10+/CD19+/CD20+/CD23−/CD43− phenotype (Figure 1B–F), raising the possibility of MCL. Flow cytometric study of the fresh nodal tissue showed the same phenotype as the leukemic cells in the peripheral blood. Histologically, the nodal architecture was effaced by largely diffuse and focally nodular patterns of lymphocytic infiltration (Figure 2). The diffuse areas contained small atypical lymphocytes with slightly irregular nuclear contours and a pale cytoplasm. The nodular areas were composed of small and large lymphocytes without polarity. Immunohistochemical study showed that the small atypical lymphocytes were B-cells expressing CD5, CD10, CD20, bcl-2, IgM, and MNDA, but not CD3, CD23, bcl-6, IgD, IRTA1, SOX11, or cyclin-D1. The Ki-67 labeling index was low, which implies a low-grade BCL. Staining with CD35 highlighted expanded and hyperplastic follicular dendritic meshworks (Figure 2), consistent with follicular colonization. MZL was diagnosed for this nodal specimen, pending a systemic survey to determine whether it was primary or secondary. 

Bone marrow involvement was confirmed by flow cytometry using marrow aspirate. The marrow trephine had nearly 100% cellularity and was diffusely infiltrated by small atypical lymphocytes expressing CD5 (weakly), CD20, and MNDA but not CD23. Subsequent abdominal CT scans showed marked splenomegaly (Figure 3A) and multiple sites of abdominal and pelvic lymphadenopathy, indicating SMZL with stage IV disease, leukemic change, and secondary nodal involvement.

The patient was treated with six courses of rituximab and four courses of cyclophosphamide, vincristine, and prednisone (COP), but he was intolerant to the regimen and developed a severe gastric ulcer with bleeding. Subsequently, the treatment was switched to bendamustine and rituximab (BR; six courses). Abdominal CT scans in November 2023 showed a significant shrinkage of the lymphadenopathy but only a partial shrinkage of the splenomegaly (from 18.6 cm to 14.6 cm; Figure 3B). He was evaluated to have achieved a partial response as of January 2024, 10 months after initial diagnosis. Although a partial shrinkage of splenomegaly may indicate residual disease, long-term management to minimize toxicity is often the goal for patients with such indolent lymphomas. Due to the drawbacks of splenectomy—short-term (perioperative events) and long-term (immune suppression and infections) complications—our clinical team chose active monitoring after discussion with the patient.

In this case, a nodal low-grade BCL with expression of CD5 and CD10 may raise the differential diagnoses of CLL/SLL, cFL, and MCL among other lymphoma types. Histologically, the infiltrate was mainly diffuse without the typical pseudofollicles or proliferation centers characteristic of SLL. Together with the absence of CD23 expression, CLL/SLL was excluded. MCL is typically CD5-positive, and the leukemic cells in the peripheral blood usually show irregular nuclear contours, so-called buttock cells [1]. In our case, negativity for SOX11 and cyclin D1 and the presence of atypical lymphocytes with hairy cytoplasmic projections make MCL unlikely. The differential diagnosis of cFL was excluded by the presence of hyperplastic/expanded follicular dendritic meshworks by CD35 staining and the absence of bcl-6 expression. Finally, a diagnosis of MZL was reached.

Jaso JM et al. characterized seven cases of CD5-positive NMZL over a 10-year interval from MD Anderson Cancer Center [3]. They found that CD5 expression in NMZL correlated with a higher frequency of dissemination as compared to CD5-negative NMZL (n = 66), but patients had an indolent clinical course and excellent overall survival similar to CD5-negative cases [3]. All cases investigated were negative for CD10 (n = 5; either by flow cytometry or immunohistochemistry), suggesting that the co-expression of CD5 and CD10 is extremely rare, if not existent, in NMZL.

CD10 is consistently expressed in FL and is rarely expressed in MALT lymphoma. In a flow cytometric study of 42 cases of small BCLs other than FL, Xu Y. et al. identified 2 CD10-positive cases, including one MCL co-expressing CD5 and one conjunctival MALT lymphoma negative for CD5 [2]. Wong E. et al. reported a rare case of nodal MZL with bright CD10 expression, but not CD5, which posed significant diagnostic challenges with FL [4].

SMZL is a rare and indolent mature BCL presenting primarily in the spleen and accounting for approximately 1% of all lymphomas. The splenomegaly is usually marked; typically >400 g and often >2 kg; frequently accompanied by circulating atypical “villous lymphocytes”; and was thus previously known as splenic lymphoma with villous lymphocytes [1,5,6]. Histologically, the spleen in SMZL is characterized by a biphasic growth pattern, comprising an inner zone of small lymphocytes and a peripheral (marginal) zone of larger lymphoid cells [5]. Usually, the splenic lymph nodes and bone marrow are also involved by a vaguely nodular infiltrate of similar nature. Immunophenotypically, tumor cells have a mature B-cell phenotype and frequently express IgM and IgD but typically lack CD5 and CD10 [5].

Establishing a definitive diagnosis of SMZL traditionally requires the evaluation of a splenectomy specimen. The histologic sections would show an expansion of white pulp nodules, classically described as having a classic biphasic appearance with central cores of mature small lymphocytes and a peripheral zone of cells with round to slightly irregular nuclei, slightly more open chromatin, and more abundant cytoplasm [5,6]. Since splenectomy is rarely performed for initial diagnosis in the modern era, currently, it would be unusual to see the classic biphasic pattern of SMZL. Accordingly, we have to rely on peripheral blood and bone marrow for diagnosis. Lymphocytes with cytoplasmic projections, historically termed villous lymphocytes, may be seen, although this feature is not required for a diagnosis of SMZL [6]. In the 2022 International Consensus Classification (ICC) [7], the presence of splenomegaly and the demonstration of a clonal B-cell population with an appropriate phenotype is sufficient for a diagnosis of SMZL [7]. Essentially, similar approaches are recommended in the 5th edition of World Health Organization Classification of Hematolymphoid Neoplasms [1].

IRTA1 (immunoglobulin superfamily receptor translocation-associated 1) is a surface B-cell receptor selectively and consistently expressed by a B-cell population located underneath and within the tonsil epithelium and dome epithelium of Peyer patches (regarded as the anatomic equivalents of a marginal zone) [8]. In a large series of MZLs (n = 590), Falini B. et al. found that IRTA1 expression was restricted to extranodal (93%) and nodal (73%) MZLs, and to lymphomas with marginal zone differentiation. Extranodal marginal zone cells with the strongest IRTA1 expression were usually located adjacent to epithelia, mimicking the IRTA1 expression pattern of normal and acquired MALT [9]. Interestingly, among the MZL cases, all 21 cases of SMZL were negative, in contrast to nodal (154/210; 73%), other extranodal (307/329; 93%), and MZL-NOS (not otherwise specified; 22/30; 73%) [9]. It seems that IRTA1-negative MZL is more in favor of SMZL than of NMZL, as in our case. In that study, IRTA1 was totally negative in a total of 219 cases of cFL (grades 1/2 and 3A) [9], indicating that negative IRTA1 expression is not useful in differentiating SMZL vs. cFL.

In summary, we present a rare case of MZL presenting in the lymph node with a dual expression of CD5 and CD10. The final diagnosis of SMZL was reached by integrating the findings of a peripheral blood smear, flow cytometry, histopathology, imaging, and clinical features.

## Figures and Tables

**Figure 1 diagnostics-14-00640-f001:**
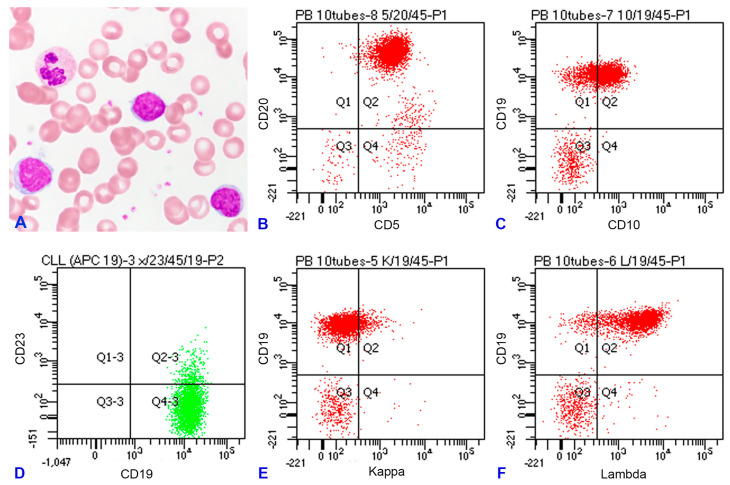
(**A**) Peripheral blood smear shows atypical lymphocytes with villous cytoplasmic projections. (**B**–**F**) Flow cytometric immunophenotyping shows that these atypical lymphocytes are B-cells expressing CD5 (dim), CD10 (dim and partial), CD19 (bright), and CD20 (bright), but not CD23. They are monotypic for lambda light chain expression.

**Figure 2 diagnostics-14-00640-f002:**
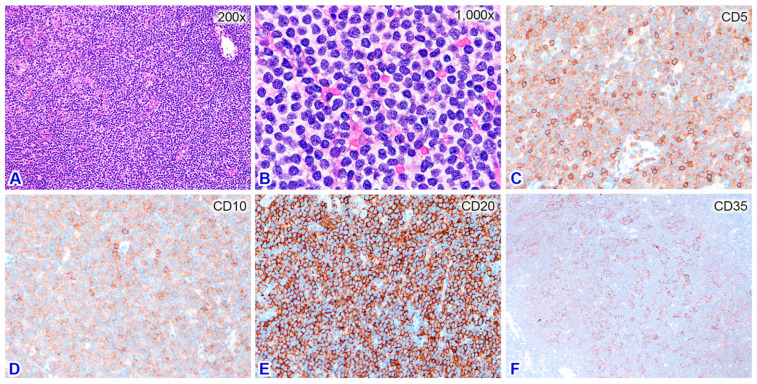
Nodal biopsy shows mainly a diffuse atypical lymphocytic infiltration ((**A**) 200×), comprising small-to-medium-sized atypical lymphocytes with slightly irregular nuclear contours and clear cytoplasm ((**B**) 1000×). Immunohistochemical study shows that the atypical lymphocytes are positive for CD5 ((**C**) 400×), CD10 ((**D**) 400×), and CD20 ((**E**) 400×). Staining with CD35 highlights expanded follicular dendritic meshworks ((**F**) 100×).

**Figure 3 diagnostics-14-00640-f003:**
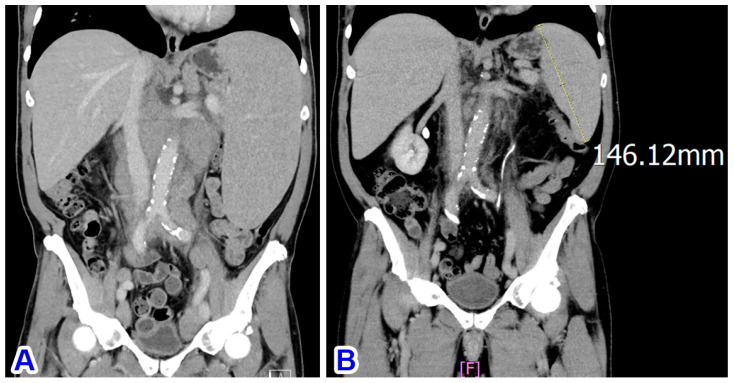
(**A**) CT scan at diagnosis shows marked splenomegaly. (**B**) Partial regression of the splenomegaly after immunochemotherapy.

## Data Availability

Data are available on request due to all institutional restrictions related to patient privacy.

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
