# Peer review of "Nodal Low-Grade B-Cell Lymphoma Co-Expressing CD5 and CD10 but Not CD23, IRTA1, or Cyclin D1: The Diagnostic Challenge of a Splenic Marginal Zone Lymphoma"

_diagnostics, 2024, doi:10.3390/diagnostics14060640_

Round 1

Reviewer 1 Report

Comments and Suggestions for Authors

Comments

1.     Regarding CT scans of the abdomen, the year 2013 is stated; however, the case presentation refers to March 2023. It is necessary to settle this disparity.

2.In addition to description of the cervical masses, additional information regarding any related symptoms could improve the clinical presentation.

3.     I suggest checking the accuracy of the values (e.g. WBC count at 51.4 x109/μL).

4.     Consider on managing the data analysis by providing each parameter explanation in a more organized way.

5.     In figure 1 the letter A is missing next to the figure.

6.     Although the Declaration of Helsinki is mentioned, obtaining the patient's informed consent only appears in the supplementary information. In case studies, patient consent and ethical issues are important and should be mentioned in the text.

7.     For flow cytometry data, it would be beneficial to include an explanation of each marker's importance in relation to a potential MCL diagnosis. Highlight the key indicators that help distinguish MZL from other lymphomas and provide a more detailed explanation.

8.     Explain the meaning of MZL and MCL when they first appear in the text.

9.     It is important to discuss the importance of a low Ki-67 marker index in the context of MZL. Explain how this finding improves our understanding of the disease.

10.  It would be beneficial for readers to mention the clinical implications of partial shrinkage of splenomegaly.

11.  Integrating IRTA1 expression into the main topic would strengthen the impact of the conclusion.

Comments on the Quality of English Language

I consider a proofreading by a certified translator is required.

Reviewer 2 Report

Comments and Suggestions for Authors

This case report presented a rare case of SMZL co-expressing CD5 and CD10 which has educational value for differential diagnosis of small B-cell disorders. Did the author obtain the cytogenetic data? It will be better to present cytogenetic data.

Two minor mistake are the parentheses in line 32 and the CT scan time in line 88. 
